# Mathematics and Numerosity but Not Visuo-Spatial Working Memory Correlate with Mathematical Anxiety in Adults

**DOI:** 10.3390/brainsci12040422

**Published:** 2022-03-22

**Authors:** Paula A. Maldonado Moscoso, Elisa Castaldi, Roberto Arrighi, Caterina Primi, Camilla Caponi, Salvatore Buonincontro, Francesca Bolognini, Giovanni Anobile

**Affiliations:** 1Institute of Neuroscience CNR, 50124 Pisa, Italy; 2Department of Neuroscience, Psychology, Pharmacology and Child Health, University of Florence, 50135 Florence, Italy; elisa.castaldi@unifi.it (E.C.); roberto.arrighi@unifi.it (R.A.); caterina.primi@unifi.it (C.P.); camilla.caponi@unifi.it (C.C.); salvatore.buonincontro@stud.unifi.it (S.B.); francesca.bolognini@stud.unifi.it (F.B.); giovanni.anobile@unifi.it (G.A.)

**Keywords:** approximate number system (ANS), math anxiety, math abilities, calculation, visuo-spatial working memory

## Abstract

Many individuals, when faced with mathematical tasks or situations requiring arithmetic skills, experience exaggerated levels of anxiety. Mathematical anxiety (MA), in addition to causing discomfort, can lead to avoidance behaviors and then to underachievement. However, the factors inducing MA and how MA deploys its detrimental effects are still largely debated. There is evidence suggesting that MA affects working memory capacity by further diminishing its limited processing resources. An alternative account postulates that MA originates from a coarse early numerical cognition capacity, the perception of numerosity. In the current study, we measured MA, math abilities, numerosity perception and visuo-spatial working memory (VSWM) in a sample of neurotypical adults. Correlational analyses confirmed previous studies showing that high MA was associated with lower math scores and worse numerosity estimation precision. Conversely, MA turned out to be unrelated to VSWM capacities. Finally, partial correlations revealed that MA fully accounted for the relationship between numerosity estimation precision and math abilities, suggesting a key role for MA as a mediating factor between these two domains.

## 1. Introduction

Emotions and feelings experienced when dealing with mathematical tasks significantly vary between individuals. For some people, mathematical tasks are a pleasant form of self-challenge; for others they represent a source of moderate but still functionally helpful anxiety. However, in some individuals, it creates excessive negative anxiety limiting performance and determining active avoidance for math-related situations [1]. Given the importance of mathematical abilities in everyday life, anxiety and avoidance for math can limit school achievement and career prospects especially in the STEM fields (science, technology, engineering, and mathematics) [2,3] and also pose challenges in many everyday activities [4].

Operationally, math anxiety (MA) has been defined as the fear and worry related to math stimuli and math-related situations [5,6]. Despite a large scientific interest in this topic, the processes underlying MA are still largely debated. Interestingly, although several studies found that individuals with high MA perform worse on math compared to those having lower MA, much evidence finds no link between MA and math ability [7]: poor math abilities seem to be insufficient and unnecessary for the development of high MA [1,8,9,10,11], with similar MA levels having been reported in students with both low and average levels of math ability [1]. Therefore, which factors determine the link between MA and math abilities? Despite studies suggesting that several environmental [12,13,14], cognitive [15,16] and genetic factors [13,14,17] might synergically interact to modulate math anxiety levels, two leading but not mutually exclusive theories (the disruption account and the reduced competency account) point to specific cognitive and perceptual factors.

The disruption account suggests that the negative impact of MA on math ability might originate from the interference between MA and working memory (WM). Worries and ruminations about math would disrupt WM resources necessary to succeed in mathematics [15]. WM is indeed a limited-capacity system enabling verbal and visuo-spatial information to be temporarily stored and manipulated [18]. However, it is not clear which component of the WM, if any, would be related to MA or whether WM would have, in some cases, a role in preventing anxiety-driven deterioration of math performance. Some studies suggested that individuals with MA present limited visuo-spatial WM resources (VSWM; [19]) while others found significant correlations between MA and the verbal component of WM [20,21]. Moreover, it is not even clear whether WM plays a role in determining the interaction or the relationship between MA and math performance. While some studies reported that better WM allows individuals to master mathematical performance in spite of high math anxiety [15,19], other studies showed the opposite pattern of results with individuals with higher WM being more prone to math failures caused by anxiety [11,22,23,24]. Finally, a recent meta-analysis questioned whether WM would play a role at all in mediating the relationship between MA and math performance [25].

The second influential theory, the reduced competency account, holds that MA might represent a by-product of poor early math performance. Maloney and colleagues [16,26,27,28] suggested that having low numerical/spatial skills, might compromise the successful development of mathematical strategies, subsequently leading to the development of MA. It has been proposed that one of the earliest signs of math performance, already present only a few hours after birth [29,30,31], is the ability to perceive non-symbolic quantities (numerosity; [32]). Some studies found that the precision of numerosity perception (also called numerical acuity) correlates with math performance, with individuals with better math skills also performing more precisely in the numerosity tasks [33,34,35,36,37,38,39,40,41,42,43,44]. It has been proposed that impairments in this early numerosity system (often named the approximate number system—ANS) might compromise the development of mathematical abilities and generate math avoidance behaviors or excessive MA [45]. However, the validity of this conclusion is still under debate: first, some studies failed to find a correlation between numerosity acuity and formal math development [46,47,48,49,50,51]. Second, the few studies investigating the relation between MA and ANS, as well as the role of MA in the relationship between ANS and math performance, came to different conclusions [14,45,52,53,54,55,56,57,58,59,60,61,62]. Some of these reports found poorer numerosity acuity in individuals with high compared to low math anxiety, a result in line with the reduced competency account [45,56,61], but other reports failed to find a significant link between ANS and MA [14,53,54,58,59,60,62]. Finally, two recent studies on adults suggested that ANS and MA might be related, with the latter serving as a mediating factor in the relationship between numerosity perception and math performance [45,52].

In addition to the heterogeneous results described above, one of the main limitations to understanding the role of MA in ANS acuity, math performance and working memory capacities is that very few studies have jointly investigated these variables in the same sample of participants. Up until today, only three studies have done so, and the results are again mixed [55,56,57]. For instance, in a study on a cohort of university students, it has been found that MA did not correlate with verbal working memory, or with ANS, but only with math performance [56]. A second study investigated the interplay between these variables in a group of adults mostly composed of individuals with ADHD or learning disabilities [55]. Results showed positive correlations between visual working memory, math abilities and MA. Numerosity perception (ANS) was instead unrelated to both math abilities and MA. Finally, Cargnelutti et al. [57] measured the interplay between these variables in 7-year-old Italian children. The results showed a positive correlation between ANS and math performance. However, math anxiety was unrelated to math performance and visual working memory.

Here, we decided to tackle the issue of the relationship between ANS, visuo-spatial working memory, MA and math performance from a quite different perspective. First, we decided not to test these domains during the developmental period or in participants with developmental disabilities as an excessive level of heterogeneity in individuals’ ability or outliers might mask significant covariation between the tested domains. Second, we measured ANS with an estimation task, previously found to be sensitive in predicting math performance [57]. Third, we measured visuo-spatial, rather than verbal, WM as there is evidence that this component might more likely be related to MA, numerosity discrimination and math performance compared to the verbal one [19,55,63,64]. To check whether and to what extent MA effects are selective to mathematics learning, we also measured, as a control learning task, participants’ reading abilities. The results showed that MA was related to both numerosity acuity and math performance. Interestingly, the correlation between numerosity precision and math performance was fully accounted for by MA levels. On the other hand, VSWM capacities were not related to MA. Overall, the present pattern of results strongly supports the idea of a close relationship between MA and the acuity of the brain mechanisms tuned to the processing of numerosity.

## 2. Materials and Methods

### 2.1. Participants

Fifty-one adults (78% female; mean age = 22.5 years, standard deviation = 7.4, range 19–54 years) with normal or corrected-to-normal vision participated in this study. Four participants did not complete the experiment and were excluded from the analysis. Participants were all psychology students in their first year of university with no mathematical or other learning disorders or overexercised calculation skills. The research was approved by the ethics committee (“Commissione per l’Etica della Ricerca”, University of Florence, 7 July 2020, No. 111), and informed consent was obtained from all participants before testing.

### 2.2. Measures

#### 2.2.1. Math Anxiety

Math anxiety was assessed by means of the Abbreviated Math Anxiety Scale (AMAS; [65], Italian version: [66]; Figure 1a). The test consists of 9 items measuring the anxiety level experienced by students in mathematical learning and testing conditions. Each item describes a different potentially anxious experience related to math (for example, “Listening to another student explaining a math formula” or “Starting a new math book chapter”). The test contains two subscales measuring math anxiety related to math evaluation (math anxiety evaluation, 4 items) and to math learning (math anxiety learning, 5 items) conditions. Participants were required to estimate how anxious they would feel during the described math-related events using a 5-point scale (ranging from “strongly agree” to “strongly disagree”). The sum of the scores based on participants’ ratings on each statement of the subscales provides a single composite score. High scores indicate high math anxiety. For the current sample, Cronbach’s α was 0.90 (IC: 0.85–0.94).

#### 2.2.2. Mathematical Abilities

Formal mathematical performance was measured by means of four different tests: the Mathematics Prerequisites for Psychometrics (MPP; [67]; Figure 1b) was used to evaluate mathematical knowledge; the Probabilistic Reasoning Scale (PRS; [68]; Figure 1) was used to evaluate probabilistic reasoning as well as simple and complex mental calculation abilities.

The Mathematics Prerequisites for Psychometrics (MPP; [67]; Figure 1b) is a questionnaire composed of 30 multiple-choice items (one correct response out of four options) evaluating the basic mathematical knowledge necessary to successfully complete the introductory statistics courses (i.e., ability to master addition, subtraction, multiplication, division with fractions and exponentiation; the set-theory principles (the branch of mathematical logic that studies sets, which can be informally described as collections of objects); fractions and decimal numbers; first-order equations; order relations between numbers from −1 to 1 (e.g., the value 0.05 is (1) lower than 0; (2) higher than 0.1; (3) within –1 and 0; (4) within 0 and 1); the concept of absolute value and the basics of probability were also included). The number of correct responses was calculated and provided a measure of the student’s math knowledge [66]. In the present sample, Cronbach’s α was 0.69 (IC: 0.65–0.80).

The Probabilistic Reasoning Scale (PRS; [68]; Figure 1b) is a 16-item questionnaire measuring basic and conditional probabilities (e.g., “A ball was drawn from a bag containing 10 red, 30 white, 20 blue, and 15 yellow balls. What is the probability that it is neither red nor blue?” Response options: (1) 30/75; (2) 10/75; (3) 45/75; the correct response is 45/75) and reasoning about random sequences of events (e.g., “A fair coin is tossed nine times. Which of the following sequence of outcomes is a more likely result of nine flips of the fair coin? (H: head, T: tail)” Response options: (1) THHTHTTHH; (2) HTHTHTHTH; (3) Both sequences are equally likely). The number of correct responses was summed and provided the probabilistic reasoning score. For the current sample, Cronbach’s alpha was 0.46 (IC: 0.22–0.65).

Finally, mental calculation abilities were measured by two custom-made computerized tests requiring participants to mentally solve simple or complex arithmetic operations (Figure 1b). Each trial started with a central fixation cross. As soon as the experimenter pressed the space bar, the stimuli (two 1° × 1.5° digits and one 1° × 1° operand, Arial font) were displayed until the participant’s response. The experimenter (blind to the stimuli) hit the spacebar as soon as the participants spelled out the result (thereby recording the response time) and then entered the response on the numeric keypad. In the simple calculation test, participants solved one-digit additions, subtractions and multiplications. In the complex calculation task, participants performed two- or three-digit additions, subtractions, multiplications and divisions. In both cases, there was no explicit time limit. None of the operations included numbers with zero (e.g., 30) or numbers with the same digits (e.g., 77). In the simple calculation task, participants performed sums of two (e.g., 2 + 5), three (e.g., 3 + 2 + 1) or four digits (e.g., 4 + 4 + 2 + 1); multiplications between two digits (e.g., 3 × 4) and subtractions between two digits (e.g., 8 − 3), solving 14 items for each operation type. For the operations between two digits, we used numbers from 2 to 9, while in the sums between three and four digits we used digits from 1 to 4, so that the calculation results were always lower than 20. Each operation was randomly selected trial-by-trial from a list of 70 operations. Response times (RTs) higher than 3 standard deviations were considered outliers and eliminated from the analysis (1.4% of trials). For the current sample, Cronbach’s α was 0.86 (IC: 0.80–0.91). During the complex calculation task, participant performed 96 trials in which they were tested with 24 subtractions, sums, multiplications and divisions. Operations between consecutive (e.g., 12 + 13, 28 − 27) or same (e.g., 17 + 17) numbers were not included. Sums and subtractions contained operations that required none, 1 or 2 carries/borrows. In the first half of the trials, operations included at least one two-digit operand, while in the second half of the task, operations included at least one three-digit operand. In the present sample, Cronbach’s α was 0.89 (IC: 0.84–0.93). For both simple and complex calculation tasks, we measured individual participants’ accuracy and average reaction time (RT), which were then transformed into z-scores. We also computed a combined index averaging the two z-scores.

#### 2.2.3. Numerosity Estimation Abilities

The proficiency of the approximate number system was measured by a numerosity estimation task. The stimuli were arrays of white squares (0.4° × 0.4°) with black borders, (to balance overall luminance; Figure 1c). On every trial, items were randomly displayed within 106 possible locations covering a 6° × 6° squared area. Each trial started with a black central fixation point that turned white after 1 s and remained on screen for the entire experiment. After 1 s, an array of small white squares was displayed around the center of the monitor for 500 ms, followed by a blank screen. Participants were asked to verbally estimate the numerosity of the set as quickly and accurately as possible. The experimenter (blind to the stimuli) hit the spacebar as soon as the response was spelled out, then entered the response on the numeric keypad and initiated the following trial (after a pause of 500 ms) by pressing the enter button. Numerosities from 5 to 12 were randomly displayed on every trial. Each participant completed 150 trials, with each numerosity presented 9 times on average. Trials with response times and responses higher than 2.5 standard deviations were considered outliers and eliminated from the analysis (2% of the trails). For the current sample, Cronbach’s α was 0.81 (IC: 0.71–0.88).

#### 2.2.4. Visuo-Spatial Working Memory

We measured visuo-spatial working memory (VSWM) by means of a computerized task (Figure 1d) inspired by the Corsi block tapping test [69]. For every trial, a fixation point was displayed on the top center of the screen with nine red squares (3° × 3°) scattered around the screen area. After 500 ms, one square at a time changed color to yellow following a given sequence (the inter stimulus interval (ISI) between color changes was 1 s) and participants were asked to repeat the sequence by tapping on the squares either in the same (forward condition) or in the opposite order (backward condition) with the two conditions tested in separate blocks of trials. Participants performed a practice trial (sequence of two squares) to become familiar with the experimental procedure, and then the task started with sequences of three squares. The sequence length was increased by one square if the participants correctly recalled at least one out of two sequences of the same length; otherwise, the test was terminated, and the span determined as the number of steps correctly reproduced. The forward and backward condition had a Cronbach’s α of 0.69 (IC: 0.51–0.81); and 0.60, (IC: 0.38–0.76), respectively, for the present sample.

#### 2.2.5. Reading Abilities

Participants were asked to read aloud 4 lists of 28 words and 3 lists of 16 non-words as fast and accurately as possible (lists taken from the Developmental Dyslexia and Dysorthography Battery 2 [70]). Reading speed was measured for each list in syllables/s, while reading accuracy was measured as the number of errors. The experimenter presented one list of words/non-words at a time. The list remained covered until the experimenter, having ascertained that the reader was ready, gave the command to GO, uncovered the list and, simultaneously, started the stopwatch. The experimenter accurately measured the reading time for each list and noted down reading errors/omissions, if present.

### 2.3. Procedure

Participants were tested in two separate sessions in a quiet room. In one session, we administered the pencil-and-paper scales (AMAS, MPP, PRS and reading test); in the second session, participants were tested with the computerized tasks (mathematics, numerosity estimation, VSWM tasks). For the computerized tasks, participants sat in front of a LG 27 monitor subtending 56° by 32° from the subject’s viewing distance of 57 cm. The monitor resolution was 1920 × 1080, and the refresh rate was 60 Hz. Stimuli were all generated and presented with Psychtoolbox [71] routines for MATLAB (ver. 2010a, The MathWorks, Inc., Natick, MA, USA).

### 2.4. Statistical Analysis

For the numerosity estimation task, we calculated the average perceived numerosity and standard deviation for each numerosity and participant separately. Precision in the estimation task was indexed in terms of Weber fractions (Wfs) calculated as the ratio between the standard deviation and the average value of the response distribution with high values of Wfs indicating low precision and vice versa. For each participant, Wfs were calculated separately for each numerosity and then averaged across numerosity levels, to obtain a comprehensive precision index. Participants’ scores were all transformed into a z-score, using the mean and standard deviation of the entire sample. Z-scores for each mathematical measure (MPP, PRS and simple and complex calculation tasks) were averaged to obtain a combined index (formal mathematics performance) that summarized the participants’ math skills. The same procedure was followed to obtain a single VSWM span value and a reading ability value.

As VSWM scores strongly deviated from normality (w = 0.9, *p* = 0.001), the relation between variables was determined by non-parametric Spearman’s zero-order and partial correlations. Log10 Bayes factors (LBFs) were also reported when appropriate. LBF values are conventionally interpreted as providing substantial (0.5–1), strong (1–2) or decisive (>2) evidence in favor of the alternative hypothesis (H1), while negative LBF within these ranges are considered as evidence for the null hypothesis (H0) [72,73].

Statistical analyses were performed using Jasp (version 0.14.1, The JASP Team 2020, https://jasp-stats.org; accessed on: 16 November 2021), MATLAB (version R2016b, The MathWorks, Inc., http://mathworks.com, accessed on: 15 September 2016) and IBM SPSS Statistics for Macintosh (version 27).

## 3. Results

To investigate whether and to what extent inter-individual differences in math anxiety levels were predicted by mathematical, visuo-spatial working memory and numerosity skills, we tested adults with several cognitive and psychophysical tasks.

For all tasks, the average scores (Table 1) were within the expected range based on standardized measures and previous studies (AMAS: 23.2, SD: 5.8 [65]; MPP: 22.8, SD: 4.56 [68]; PRS: 12.73, SD: 2.59 [68]; numerosity Wf: 0.097, SD: 0.01 (22–32 y.o years old; [74]); Corsi span forward: 6.0, SD: 1.09; Corsi span backward: 5.24, SD: 0.90 (20–30 years old; [75]); word reading accuracy: 0.76, SD: 1.07; non-word reading accuracy: 1.91, SD: 1.7; word reading speed: 5.4, SD: 0.93; non-word reading speed: 3.27, SD: 0.7 (adults; [76])).

Given that all the mathematical tasks turned out to be highly correlated with each other (all rho > 0.39, all *p* < 0.009, all LBF > 1), we computed a single index to estimate the formal mathematics performance by averaging the z-scores across the tasks. We also computed a single VSWM index, given that participants’ span in the forward and backward condition highly correlated with each other (rho = 0.43, *p* = 0.002, LBF = 2.2), and for the same reason, we calculated a single reading index (word reading performance and non-word reading performance: rho = 0.53, *p* = 0.0001, LBF = 2.1).

The results, depicted in Figure 2a (see also Table 2) clearly showed that participants with higher MA levels were also those showing lower formal math performance (rho = –0.44, *p* = 0.002, Bonferroni corrected α = 0.005, LBF = 1.1). We also found that the correlation between MA and reading index was not statistically significant (rho = 0.19, *p* = 0.2, LBF = –0.43; Bonferroni corrected α = 0.005; Table 2), suggesting that MA does not act as a general predictor of learning abilities but that it is specifically linked to math.

Since the main goal of this study was to investigate the interplay between MA levels with mathematical performance, numerosity perception and VSWM, we cross-correlated these variables (Table 2 and Figure 2).

We found that individuals with higher MA have lower numerosity acuity (higher Wf; rho = 0.48, *p* = 0.0006, LBF = 2.1; Bonferroni corrected α = 0.005). Crucially, for the purpose of the current study, the performance in the VSWM task was unrelated to MA levels (rho = –0.09, *p* = 0.55, Bonferroni corrected α = 0.005). This null correlation was clearly supported by a Bayesian non-parametric analysis showing substantial evidence in favor of the null hypothesis (LBF = –0.7). We did not find any significant correlation between MA levels and VSWM, even when analyzing the performance in the forward and backward VSWM conditions separately (forward: rho = –0.25, *p* = 0.09, LBF = –0.02; backward: rho = 0.1, *p* = 0.49, LBF = –0.6; Bonferroni corrected α = 0.005).

Since between-subject variability is a fundamental prerequisite for correlations, the fact that ANS but not VSWM correlated with MA could reflect a statistical artifact due to a potentially lower variability in VSWM scores. To rule out this possibility, we performed a bootstrap sign test on task variance. At each bootstrap iteration (10,000 iterations), we independently resampled (with replacement, as many indices as the number of participants) participants’ Wfs and VSWM z-scores and calculated the between-subject variance for the two tasks. We than computed the p-value as the proportion of times the Wf variance was higher compared to that provided by the VSWM task. The p-value was 0.44, indicating that these tasks had a similar variability level, suggesting that the different pattern of correlation with MA was unlikely due to a difference in variability levels.

Figure 3 shows that, when controlling for MA, the correlation between formal math performance and numerosity Wf was not statistically significant (rho = –0.28, *p* = 0.06), suggesting that MA played a crucial role in driving this correlation. The correlation between math anxiety and formal math performance scores was not statistically significant when numerosity Wf was controlled as a covariate (rho = –0.29, *p* = 0.05), suggesting, on the other hand, that numerosity Wf also had a role in the relationship between MA and math performance. When controlling for formal math performance scores, the correlation between numerosity Wf and math anxiety remained statistically significant (rho = 0.36, *p* = 0.014), indicating that formal math performance was not sufficient to fully account for the correlation between numerosity perception and MA levels.

## 4. Discussion

In the current study, we investigated the role of domain-general (visuo-spatial working memory, VSWM) and domain-specific (numerosity acuity) factors in determining math anxiety (MA) levels as well as its relation to formal math performance. Preliminarily, we tested whether MA was specifically related to formal math performance or also to other school domains, such as reading. The results showed that MA was specifically linked to formal math and not to reading abilities. Even more importantly, MA was related to numerosity acuity and independent from VSWM, with the link between numerosity acuity and math performance being fully accounted for by MA. Moreover, we found that numerosity acuity played a role in driving the relationship between MA and formal math performance. Overall, the work reported here suggests that adults with higher levels of math anxiety also have lower math performance and a noisier sense of number (higher Wf), in line with the reduced competency account.

In line with this theoretical framework, we found here that MA and numerosity acuity were significantly correlated; that is, individuals with higher math anxiety levels also showed higher Wf (lower ANS precision). This result nicely complements a previous study reporting that individuals with high math anxiety showed a lower accuracy (correct responses) in a numerosity discrimination task compared to their peers with lower levels of math anxiety [45]. Here, we quantified the sensitivity of the ANS by measuring Wf (rather than proportion of correct responses) and, therefore, considered, more appropriately, the sensory precision of the system by also generalizing the previous reports to a different paradigm (numerosity estimation rather than discrimination). However, it is worth noting that another study failed to find a significant correlation between MA and ANS in adults [56] despite ANS acuity being measured as in Lindskog et al. [45]. Braham and Libertus [56] suggested that a possible explanation for this discrepancy might rely on the lower variability of scores in the math anxiety questionnaire they obtained relative to those reported by Lindskog et al. (24% vs. 66%). Here, we found a significant correlation between MA and ANS despite the fact that the standard deviation of anxiety scores measure in our study (28% of the mean) was similar to that in Braham and Libertus. Given that the variability in the MA scores does not appear to be a crucial factor in determining the correlation between ANS acuity and MA, the differences across studies might be related to the methods used to measure ANS acuity. Using an estimation task might provide more reliable measures of ANS precision and allow for the detection of the correlation between this variable and MA even for low variability in the MA scores.

The present results also support previous findings showing that adults with higher approximate number system acuity are also those with higher math performance ([43,45,56,77]; for a meta-analysis, see [78]; although, see [40,79,80], for a different account). Importantly, in the current study, we found that the correlation between ANS and math performance turned out to be not statistically significant when controlling for MA. This is in line with two recent studies finding that MA fully accounts for the relationship between ANS acuity and math performance. For example, Lindskog et al. [45] reported a significant mediation role of MA in the link between ANS and math performance. Similarly, Maldonado Moscoso et al. [52] found a mediatory role of MA in determining the relationship between ANS and math proficiency by taking into account MA in individuals with high MA. We also found that the correlation between MA and formal math performance was fully accounted for by numerosity acuity. Taken together, these results support the hypothesis that individuals with high MA may have a coarser ANS acuity [45], in line with the reduced competency account. Having a poor ANS during development could increase the number of negative experiences related to math learning, increasing the probability of developing MA. In turn, MA impedes performance, bringing more anxiety and avoidance behavior.

In the current study, we did not find a significant relationship between MA and WM resources. That domain-general functions do not covary with MA is in line with a previous study reporting a no significant correlation between MA and visuo-spatial attention [52]. However, this is in contrast with several previous studies indicating that higher levels of MA were associated with poor WM performance [15,81,82]. A possible explanation for the lack of correlation between WM and MA reported here, compared to other studies [83], might be the kind of WM taken into account as well as the task employed to measure WM. As suggested by Namkung and colleagues [84], math anxiety might be prompted most strongly by those WM tasks that involve the manipulation of numerical information. For instance, Ashcraft and Kirk [15] found that MA correlated with working memory only when the task used to measure WM involved arithmetic or math-related stimuli (computation-based working memory) but not verbal stimuli. Similar results were found by other groups that used computation-based WM [23,85,86]. Here, we tested participants’ VSWM using a number-free task and did not find a significant relationship between MA and VSWM, as well as between VSWM and math performance. Therefore, one possibility is that only some “domain-specific” components of WM, potentially those strongly related to math concepts, might be relevant to MA and to its relationship with math performance. However, other factors should be considered as well. Indeed, previous studies that have measured VSWM did find a relationship between this WM component and MA (with individuals with higher levels of MA having poorer VSWM resources [19,63]). Nevertheless, the types of tasks used in these studies to measure VSWM were different compared to ours. Georges et al. [63] used the no-grid protocol taken from a grid/no-grid task, which required participants to report whether a comparison configuration was in accordance or not with the spatial locations of target crosses, while Miller et al. [19] used a paper-folding task. These tasks might require different cognitive abilities (for example mental rotation or visual imagery) more related to MA than those involved in our task. Moreover, the test used to measure MA as well as the educational background of the participant tested in those studies differed compared to ours, potentially explaining the discrepancy between the current and previous results. Finally, the strategies applied by participants (i.e., spatializing verbal sequences in mind) to solve WM tasks might also explain the correlation with MA [87]. In order to test for these possibilities, future studies should measure different types of WM, with different tasks, and investigate their specific relationship with MA in the same participants.

The strong negative correlation between MA levels and mathematical performance reported here is in line with several previous findings suggesting that repeated experiences of failures in mathematics-related situations may generate anxious feelings when dealing with math tests ([7,9,11,15,28,45,88,89,90,91,92,93]; for a review, see [94,95]; for a meta-analysis, see [1,84,96,97]). Moreover, the current results make clear that the negative effects of math anxiety only affect math performance and leave other school-based abilities unaffected. Indeed, we found that MA specifically predicted math performance but not reading abilities, suggesting that MA is not a general predictor of learning [17]. In the current study, we did not measure individuals’ general academic anxiety, so it is not possible to definitively exclude the possibility that this does not also reflect a generalized state of non-specific anxiety. However, previous studies have tackled this question and showed that math performance specifically correlated with MA and not with test or performance anxiety ([45,52,98]; for a meta-analysis, see [1]). Nevertheless, although we cannot demonstrate that the test used here specifically measured MA, as opposed to general performance anxiety, the fact that MA did not correlate with reading abilities makes this possibility unlikely since there is no reason to believe that general anxiety would impact math more than other academic skills. This finding is also interesting, as a previous study in 7-year-old Italian children found that only general, and not math-specific anxiety, predicted math performance [57]. The fact that we found it here in a sample of adults suggests that the specificity of this link might develop after prolonged experience with math education.

The current study also has some limitations. Mathematical abilities are heterogeneous and involve several different competencies such as counting, mental calculation, written calculation, verbal math knowledge, number reading and many others. In this study, we estimated mathematical proficiency via a multidimensional composite index, but obviously it did not entail all math-related abilities. The correlation patterns described in this study cannot, therefore, be generalized to all mathematical sub-competencies, an important issue requiring future studies to be fully addressed. Another limitation is related to the selected sample. The current study describes the interaction between MA, ANS and math competency in adults, but these results cannot be easily generalized to children or adolescents. Again, future studies (ideally leveraging on a longitudinal approach) are needed to tackle this issue directly and complement the present results with a developmental trajectory of the relationship between MA, ANS and math capacities. Finally, in the current experiment, we did not balance the ratio between female and male participants, a decision mainly driven by recent meta-analyses showing no gender effect on the association between math achievement and MA [25,96]. However, this is nevertheless a limitation, and the current results need to be replicated with balanced samples before being fully generalizable.

## 5. Conclusions

Taken together, our results showed that individuals with high MA also have poor ANS and worse math performance. During development, a poor ANS could increase the likelihood of initial failure and negative learning experiences during math education, thereby triggering the development of MA. VSWM, on the other hand, did not seem to play a key role in determining MA. Overall, these results strongly support the reduced competency account.

## Figures and Tables

**Figure 1 brainsci-12-00422-f001:**
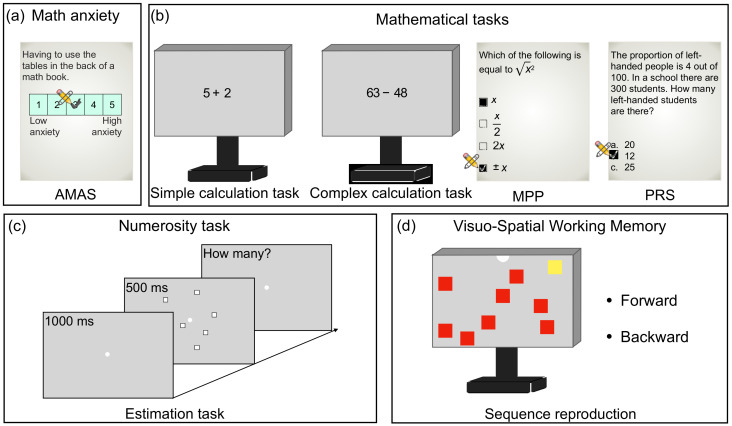
Illustration of task and stimuli. (**a**) Example of one item of the Abbreviated Math Anxiety Scale. (**b**) Mathematical tasks. We measured participants’ math performance through two computerized tests (simple and complex calculation tasks) and two paper-and-pencil questionnaires (Mathematics Prerequisites for Psychometrics—MPP, and Probabilistic Reasoning Scale—PRS). (**c**) Illustration of the numerosity estimation task. (**d**) VSWM was assessed by a computerized task. Participants observed the sequence of squares turning to yellow and then repeated the sequence in the same (forward condition) or reverse (backward condition) order.

**Figure 2 brainsci-12-00422-f002:**
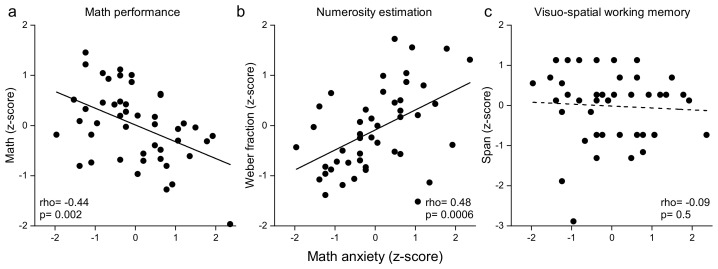
Correlations between math anxiety, formal math performance (**a**), numerosity estimation acuity (Weber fraction, (**b**)), and VSWM scores (span, (**c**)). Lines represent best linear fitting; dots represent individual participant scores. *p* < Bonferroni corrected α = 0.05/10 = 0.005.

**Figure 3 brainsci-12-00422-f003:**
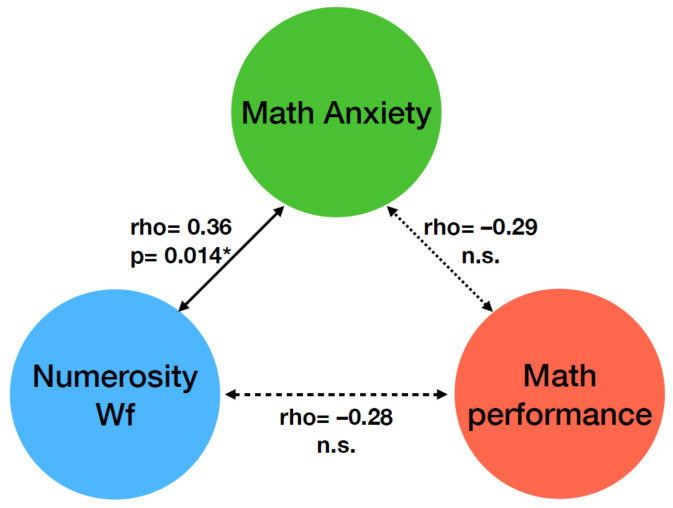
Diagrams of partial correlations between math anxiety, numerosity Wf and formal math performance. Values report partial correlations between the two variables connected by arrows after controlling for the third variable. * *p* < 0.05.

**Table 1 brainsci-12-00422-t001:** Descriptive statistics. Mean and standard deviation (SD) of participants’ performance for the various measures.

Measures	Mean	SD
Math anxiety evaluation	14.78	3.57
Math anxiety learning	9.91	4.04
Simple calculation accuracy	0.96	0.04
Simple calculation RT	1.79	0.27
Complex calculation accuracy	0.75	0.14
Complex calculation RT	15.88	6.44
Mathematics Prerequisites for Psychometrics	24.22	3.37
Probabilistic Reasoning Scale	13.67	1.90
Numerosity Wf	0.07	0.02
VSWM forward	6.22	1.17
VSWM backward	6.35	0.87
Word reading accuracy	0.35	0.64
Non-word reading accuracy	1.65	1.95
Word reading speed	5.53	0.93
Non-word reading speed	3.59	0.73

**Table 2 brainsci-12-00422-t002:** Correlational matrix. Spearman’s correlations and *p*-values between the various measures.

Variables	1	2	3	4	5
	Math Anxiety	Formal Math Performance	Numerosity Wf	VSWM	Reading Index
**1**	--				
**2**	**rho = –0.44** ***p* = 0.002**	--			
**3**	**rho = 0.48** ***p* = 0.0006**	**rho = –0.43** ***p* = 0.003**	--		
**4**	rho = –0.09 *p* = 0.55	rho = 0.14 *p* = 0.36	rho = 0.07 *p* = 0.6	--	
**5**	rho = 0.19 *p* = 0.20	rho = –0.08 *p* = 0.58	rho = 0.23 *p* = 0.12	rho = –0.17 *p* = 0.26	--

Bold numbers report statistically significant correlation after Bonferroni correction (α = 0.005).

## Data Availability

Data for the main findings are available at: https://doi.org/10.5281/zenodo.6353646 (accessed on 30 January 2022).

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
