# Peer review of "Mathematics and Numerosity but Not Visuo-Spatial Working Memory Correlate with Mathematical Anxiety in Adults"

_brainsci, 2022, doi:10.3390/brainsci12040422_

Round 1

Reviewer 1 Report

Review: Mathematics and numerosity but not visuo-spatial working memory correlate with math-anxiety in young adults

This story explores the relationship between math anxiety and other cognitive components of math performance, such as numerosity, measures of visuospatial working memory, and standard tests of math performance. Past research on the Disruption Account of math anxiety suggest that cognitive processes related to anxiety utilize finite working memory resources, therefore disrupting task-related thought processes that should be devoted to completing mathematics. In this study, the scientists evaluated the role of visuospatial working memory in math anxiety, and the relation of this factor to both math anxiety and math performance. By contrast, the Reduced Competency Account suggests that math anxious individuals may have fundamental differences in understanding underlying numerical concepts, which causes anxiety and deficits in skills that build on this foundational knowledge. In the current study, in order to measure this factor, the authors included a measure of numerosity using magnitude comparisons. The authors evaluated these two accounts, exploring whether numerosity or working memory would be account for the relationship between math anxiety and math performance. The results suggest that numerosity is related to math anxiety, and math performance. Conversely, visuospatial working memory was not related to math anxiety or math performance, contrary to results from previous studies. In addition, variance in numerosity also accounted for the relationship between math anxiety and math performance. Overall, the authors suggest that since numerosity was shown to account for more variance related to math anxiety compared to visuospatial working memory, the Reduced Competency Account may provide a more complete picture of how math anxiety relates to math performance. 

Overall, the manuscript is well-written, the study has clear connections to past research, and the analyses are clearly motivated, and easy to interpret. This was an enjoyable article to read and review, and it’s clear that the authors have put a lot of thought and effort into streamlining the explanation and results. This article provides interesting support, and I think is an important incremental step and fills in an important gap in the math anxiety literature. However, I do have some issues that I think need to be addressed. 

Big Issues:

I was a little confused about the hypotheses related to working memory. Specifically, some of the working memory literature in the past has suggested working memory is disrupted in math anxiety. However, there has been more literature that has focused on differentiating effects related to verbal and visuospatial working memory. For example, in the explanation of the Disruption Account that is used by the authors, they explain that rumination and intrusive negative thoughts may disrupt working memory. In that case, because these behaviors are more closely related to negative self-talk, we might expect that verbal working memory resources would be more closely related to the “Disruption Account.” In addition, there is also work exploring the relationship between math anxiety and spatial skills, and visuospatial working memory. Although the authors address some of this heterogeneity, I think they remained fairly agnostic to some of the variation in the literature, and I would like to know more about their perspective, and why it informed the operational choices in this study. I would appreciate more clear delineation of this literature, and why the authors specifically chose visuospatial working memory as the factor they chose to represent working memory, and why a measure of verbal working memory was not included. 

Smaller issues:

I was confused by the explanation of the Numerosity Weber Fraction (Wf). Could the authors provide more clarity (and/or an example) of how this is calculated so that the reader can have a better understanding of how to interpret this factor? 

Overall the manuscript was very well-written and easy to understand. However, there were a few instances where grammar was not quite correct (subject-verb agreement and or plurals, such as “a sixteen items questionnaires”). These were not distracting and did not detract from the high quality of the paper, but should be corrected before publication. 

The varying measures of mathematical performance all address very different components of math performance. Although the authors have addressed this to a degree, I would appreciate a little more information as to why these measures were chosen. Relatedly, this would inform why they were combined into one unifying measure of math performance by averaging the z-scores on these measures.

Overall, I think the authors have put together a simple, well-done study. The design of the study is very clear, motivated in previous literature, and does an excellent job of using appropriate statistical analysis to address the research questions. I enjoyed reading and reviewing this paper, and thank the authors for their hard work on this manuscript.

Reviewer 2 Report

The paper contrasts the disruption account and the reduced competency account for math anxiety and is framed as a test between these two hypotheses.  The goal of this study was thus to examine possible relationships between math anxiety (MA), numerosity perception, visuo-spatial working memory (VSWM), and math performance in college students. Confirming prior reports, MA was negatively correlated with math performance and numerosity perception.  The author argue the findings support the reduced competency hypothesis because MA mediates the relationship between symbolic math and numerosity perception.

Their methods differ from all of the prior studies that they review that investigated the relationship between ANS representations and math anxiety.  In the previous studies a numerosity comparison task was used.  The current study uses a mapping task where participants symbolically labeled each numerosity (ie., labeled 8 dots as “eight”).  They compute a weber fraction based on the variance in the reported numerical values.  This difference may explain the difference between the results of the Libertus study and their own study.  Libertus et al did not find a link between ANS and MA. The authors have a good discussion of the variance in MA scores and provide evidence that this is not likely to be the root of the difference between their finding and Libertus et al.

An issue that the authors should discuss is that their math and numerosity perception tasks used the unconventional approach of having an experimenter record responses.  The experimenter pressed the space bar when the participants started to respond to increase RT precision and then typed in the response.  This is sometimes done with children but rare for an adult sample and differs from their method for the reading task (more detail needed there)  We bring this up because it seems highly likely that these methods would be more likely to induce math anxiety than if participants recorded their own responses. Could the correlations be partially due then to math anxiety influencing performance in the numerosity and math tasks during testing?

MA was correlated with composite math scores but not reading scores.  Importantly, they found that when they controlled for math scores the correlation between numerosity perception and MA still held. The link between composite math scores and numerosity perception was fully accounted for by MA and the authors provide a nice discussion of how this might MA over development might lead to worse math performance.  This hypothesis should be tested longitudinally in future work.

Line 124: the participants’ age range is listed as 10-54, but the title of this paper says the target group is young adults.  The paper also says the participants were college students… 10 year-old college students?

Females in this study comprised 78% of the population. But gender differences in MA on math performance, ANS acuity were not reported in the data analysis.

Line 41 use of word trivial seems odd here not sure what it means.
